# ARE ADVERSARIAL EXAMPLES INEVITABLE?

## ABSTRACT

A wide range of defenses have been proposed to harden neural networks against adversarial attacks. However, a pattern has emerged in which the majority of adversarial defenses are quickly broken by new attacks. Given the lack of success at generating robust defenses, we are led to ask a fundamental question: Are adversarial attacks inevitable?

This paper analyzes adversarial examples from a theoretical perspective, and identifies fundamental bounds on the susceptibility of a classifier to adversarial attacks. We show that, for certain classes of problems, adversarial examples are inescapable. Using experiments, we explore the implications of theoretical guarantees for real-world problems and discuss how factors such as dimensionality and image complexity limit a classifier's robustness against adversarial examples.

## 1 INTRODUCTION

A number of adversarial attacks on neural networks have been recently proposed. To counter these attacks, a number of authors have proposed a range of defenses. However, these defenses are often quickly broken by new and revised attacks. Given the lack of success at generating robust defenses, we are led to ask a fundamental question: Are adversarial attacks inevitable?

In this paper, we identify a broad class of problems for which adversarial examples cannot be avoided. We also derive fundamental limits on the susceptibility of a classifier to adversarial attacks that depend on properties of the data distribution as well as the dimensionality of the dataset.

Adversarial examples occur when a small perturbation to an image changes its class label. There are different ways of measuring what it means for a perturbation to be "small"; as such, our analysis considers a range of different norms. While the $\ell_\infty$-norm is commonly used, adversarial examples can be crafted in any $\ell_p$-norm (see Figure 1). We will see that the choice of norm can have a dramatic effect on the strength of theoretical guarantees for the existence of adversarial examples. Our analysis also extends to the $\ell_0$-norm, which yields "sparse" adversarial examples that only perturb a small subset of image pixels (Figure 2).

| Original | $\ell_2$-norm=10 | $\ell_\infty$-norm=0.05 | $\ell_0$-norm=5000 (sparse) |

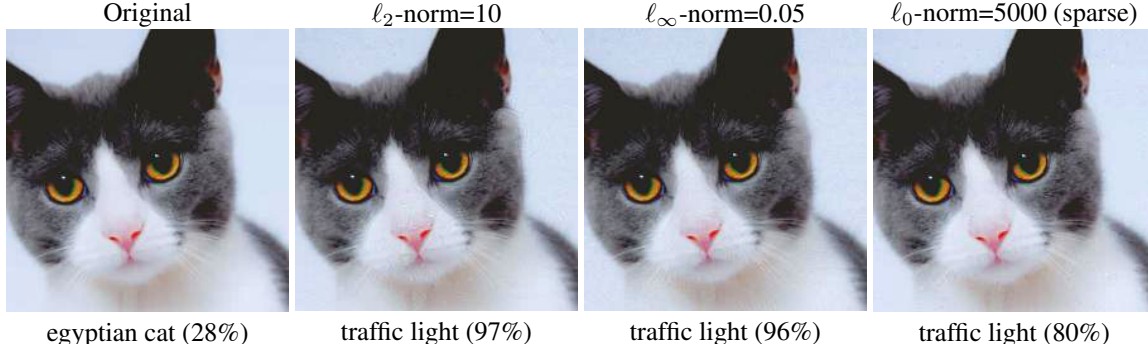

| egyptian cat (28%) | traffic light (97%) | traffic light (96%) | traffic light (80%) |

Figure 1: Adversarial examples with different norm constraints formed via the projected gradient method (Madry et al., 2017) on Resnet50, along with the distance between the base image and the adversarial example, and the top class label.

As a simple example result, consider a classification problem with $n$-dimensional images with pixels scaled between 0 and 1 (in this case images live inside the unit hypercube). If the image classes each occupy a fraction of the cube greater than $\frac{1}{2}\exp(-\pi\epsilon^2)$, then images exist that are susceptible to adversarial perturbations of $\ell_2$-norm at most $\epsilon$. Note that $\epsilon = 10$ was used in Figure 1, and larger values are typical for larger images.

Finally, in Section 8, we explore the causes of adversarial susceptibility in real datasets, and the effect of dimensionality. We present an example image class for which there is no fundamental link between dimensionality and robustness, and argue that the data distribution, and not dimensionality, is the primary cause of adversarial susceptibility.

## 1.1 BACKGROUND: A BRIEF HISTORY OF ADVERSARIAL EXAMPLES

Adversarial examples, first demonstrated in Szegedy et al. (2013) and Biggio et al. (2013), change the label of an image using small and often imperceptible perturbations to its pixels. A number of defenses have been proposed to harden networks against attacks, but historically, these defenses have been quickly broken. *Adversarial training*, one of the earliest defenses, successfully thwarted the fast gradient sign method (FGSM) (Goodfellow et al., 2014), one of the earliest and simplest attacks. However, adversarial training with FGSM examples was quickly shown to be vulnerable to more sophisticated multi-stage attacks (Kurakin et al., 2016; Tramèr et al., 2017a). More sophisticated defenses that rely on network distillation (Papernot et al., 2016b) and specialized activation functions (Zantedeschi et al., 2017) were also toppled by strong attacks (Papernot et al., 2016a; Tramèr et al., 2017b; Carlini & Wagner, 2016; 2017a).

The ongoing vulnerability of classifiers was highlighted in recent work by Athalye et al. (2018) and Athalye & Sutskever (2017) that broke an entire suite of defenses presented in ICLR 2018 including thermometer encoding (Buckman et al., 2018), detection using local intrinsic dimensionality (Ma et al., 2018), input transformations such as compression and image quilting (Guo et al., 2017), stochastic activation pruning (Dhillon et al., 2018), adding randomization at inference time (Xie et al., 2017), enhancing the confidence of image labels (Song et al., 2017), and using a generative model as a defense (Samangouei et al., 2018).

Rather than hardening classifiers to attacks, some authors have proposed sanitizing datasets to remove adversarial perturbations before classification. Approaches based on auto-encoders (Meng & Chen, 2017) and GANs (Shen et al., 2017) were broken using optimization-based attacks (Carlini & Wagner, 2017b;a).

A number of "certifiable" defense mechanisms have been developed for certain classifiers. Raghunathan et al. (2018) harden a two-layer classifier using semidefinite programming, and Sinha et al. (2018) propose a convex duality-based approach to adversarial training that works on sufficiently small adversarial perturbations with a quadratic adversarial loss. Kolter & Wong (2017) consider training a robust classifier using the convex outer adversarial polytope. All of these methods only consider robustness of the classifier on the training set, and robustness properties often fail to generalize reliably to test examples.

One place where researchers have enjoyed success is at training classifiers on low-dimensional datasets like MNIST (Madry et al., 2017; Sinha et al., 2018). The robustness achieved on more complicated datasets such

| original | $\ell_\infty$-norm | $\ell_0$-norm (sparse) | sparse perturbation |

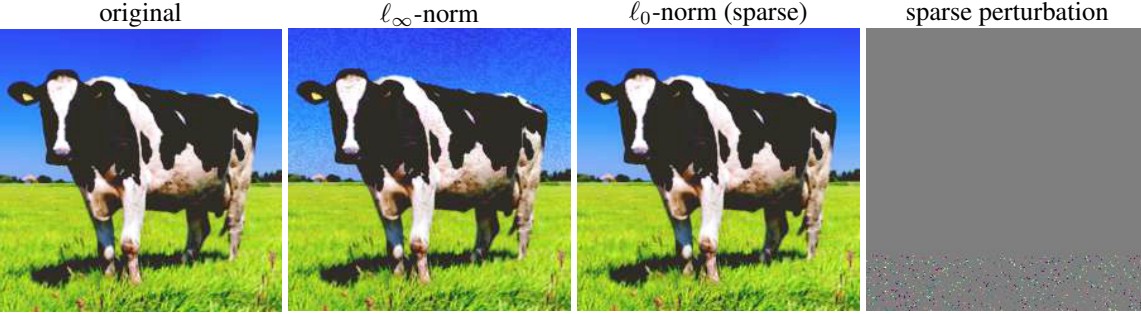

Figure 2: Sparse adversarial examples perturb a small subset of pixels and can hide adversarial "fuzz" inside high-frequency image regions. The original image (left) is classified as an "ox." Under $\ell_\infty$-norm perturbations, it is classified as "traffic light", but the perturbations visibly distort smooth regions of the image (the sky). These effects are hidden in the grass using $\ell_0$-norm (sparse) perturbations limited to a small subset of pixels.

as CIFAR-10 and ImageNet are nowhere near that of MNIST, which leads some researchers to speculate that adversarial defense is fundamentally harder in higher dimensions – an issue we address in Section 8.

This paper uses well-known results from high-dimensional geometry, specifically isoperimetric inequalities, to provide bounds on the robustness of classifiers. Several other authors have investigated adversarial susceptibility through the lens of geometry. Fawzi et al. (2018) study adversarial susceptibility of datasets under the assumption that they are produced by a generative model that maps random Gaussian vectors onto images. Gilmer et al. (2018) do a detailed case study, including empirical and theoretical results, of classifiers for a synthetic dataset that lies on two concentric spheres. Simon-Gabriel et al. (2018) show that the Lipschitz constant of untrained networks with random weights gets large in high dimensions. Shortly after the original appearance of our work, Mahloujifar et al. (2018) presented a study of adversarial susceptibility that included both evasion and poisoning attacks. Our work is distinct in that it studies adversarial robustness for arbitrary data distributions, and also that it rigorously looks at the effect of dimensionality on robustness limits.

### 1.2 NOTATION

We use $[0,1]^n$ to denote the unit hypercube in $n$ dimensions, and $\text{vol}(\mathcal{A})$ to denote the volume (i.e., n-dimensional Lebesgue measure) of a subset $\mathcal{A} \subset [0,1]^n$. We use $\mathbb{S}^{n-1} = \{x \in \mathbb{R}^n \,|\, \|x\|_2 = 1\}$ to denote the unit sphere embedded in $\mathbb{R}^n$, and $s_{n-1}$ to denote its surface area. The size of a subset $\mathcal{A} \in \mathbb{S}^{n-1}$ can be quantified by its ($n-1$ dimensional) measure $\mu[\mathcal{A}]$, which is simply the surface area the set covers. Because the surface area of the unit sphere varies greatly with $n$, it is much easier in practice to work with the *normalized measure*, which we denote $\mu_1[\mathcal{A}] = \mu[\mathcal{A}]/s_{n-1}$. This normalized measure has the property that $\mu_1[\mathbb{S}^{n-1}] = 1$, and so we can interpret $\mu_1[\mathcal{A}]$ as the probability of a uniform random point from the sphere lying in $\mathcal{A}$. When working with points on a sphere, we often use geodesic distance, which is always somewhat larger than (but comparable to) the Euclidean distance. In the cube, we measure distance between points using $\ell_p$-norms, which are denoted

$$\|z\|_p = \left( \sum_i |z_i|^p \right)^{1/p} \text{ if } p > 0, \text{ and } \|z\|_0 = \text{card}\{z_i \,|\, z_i \neq 0\}.$$

Note that $\|\cdot\|_p$ is not truly a norm for $p < 1$, but rather a semi-norm. Such metrics are still commonly used, particularly the "$\ell_0$-norm" which counts the number of non-zero entries in a vector.

## 2 PROBLEM SETUP

We consider the problem of classifying data points that lie in a space $\Omega$ (either a sphere or a hypercube) into $m$ different object classes. The $m$ object classes are defined by probability density functions $\{\rho_c\}_{c=1}^m$, where $\rho_c : \Omega \to \mathbb{R}$. A "random" point from class $c$ is a random variable with density $\rho_c$. We assume $\rho_c$ to be bounded (i.e., we don't allow delta functions or other generalized functions), and denote its upper bound by $U_c = \sup_x \rho_c(x)$.

We also consider a "classifier" function $\mathcal{C} : \Omega \to \{1, 2, \ldots, m\}$ that partitions $\Omega$ into disjoint measurable subsets, one for each class label. The classifier we consider is discrete valued – it provides a label for each data point but not a confidence level.

With this setup, we can give a formal definition of an adversarial example.

**Definition 1.** *Consider a point $x \in \Omega$ drawn from class $c$, a scalar $\epsilon > 0$, and a metric $d$. We say that $x$ admits an $\epsilon$-adversarial example in the metric $d$ if there exists a point $\hat{x} \in \Omega$ with $\mathcal{C}(\hat{x}) \neq c$, and $d(x, \hat{x}) \leq \epsilon$.*

In plain words, a point has an $\epsilon$-adversarial example if we can sneak it into a different class by moving it at most $\epsilon$ units in the distance $d$.

We consider adversarial examples with respect to different $\ell_p$-norm metrics. These metrics are written $d_p(x, \hat{x}) = \|x - \hat{x}\|_p$. A common choice is $p = \infty$, which limits the absolute change that can be made to any one pixel. However, $\ell_2$-norm and $\ell_1$-norm adversarial examples are also used, as it is frequently easier to create adversarial examples in these less restrictive metrics.

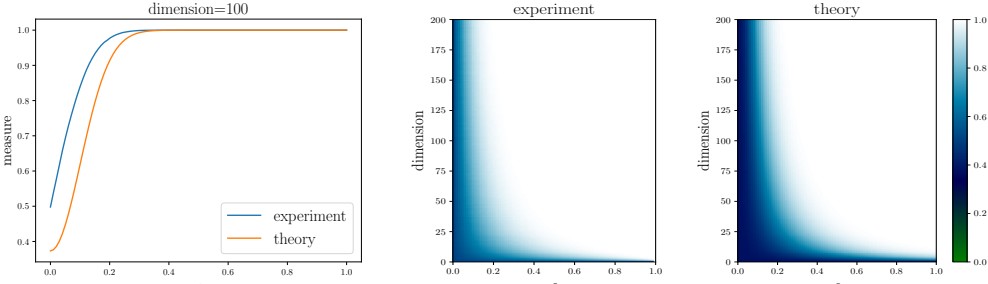

Figure 3: The $\epsilon$-expansion of a half sphere nearly covers the whole sphere for small $\epsilon$ and large $n$. Visualizations show the fraction of the sphere captured within $\epsilon$ units of a half sphere in different dimensions. Results from a near-exact experimental method are compared to the theoretical lower bound in Lemma 2.

We also consider *sparse* adversarial examples in which only a small subset of pixels are manipulated. This corresponds to the metric $d_0$, in which case the constraint $\|x - \hat{x}\|_0 \leq \epsilon$ means that an adversarial example was crafted by changing at most $\epsilon$ pixels, and leaving the others alone.

## 3   THE SIMPLE CASE: ADVERSARIAL EXAMPLES ON THE UNIT SPHERE

We begin by looking at the case of classifiers for data on the sphere. While this data model may be less relevant than the other models studied below, it provides a straightforward case where results can be proven using simple, geometric lemmas. The more realistic case of images with pixels in $[0, 1]$ will be studied in Section 4.

The idea is to show that, provided a class of data points takes up enough space, nearly every point in the class lies close to the class boundary. To show this, we begin with a simple definition.

**Definition 2.** *The $\epsilon$-expansion of a subset $\mathcal{A} \subset \Omega$ with respect to distance metric $d$, denoted $\mathcal{A}(\epsilon, d)$, contains all points that are at most $\epsilon$ units away from $\mathcal{A}$. To be precise*

$$\mathcal{A}(\epsilon, d) = \{x \in \Omega \,|\, d(x, y) \leq \epsilon \text{ for some } y \in \mathcal{A}\}.$$

*We sometimes simply write $\mathcal{A}(\epsilon)$ when the distance metric is clear from context.*

Our result provides bounds on the probability of adversarial examples that are independent of the shape of the class boundary. This independence is a simple consequence of an *isoperimetric inequality*. The classical isoperimetric inequality states that, of all closed surfaces that enclose a unit volume, the sphere has the smallest surface area. This simple fact is intuitive but famously difficult to prove. For a historical review of the isoperimetric inequality and its variants, see Osserman et al. (1978). We will use a special variant of the isoperimetric inequality first proved by Lévy & Pellegrino (1951) and simplified by Talagrand (1995).

**Lemma 1** (Isoperimetric inequality). *Consider a subset of the sphere $\mathcal{A} \subset \mathbb{S}^{n-1} \subset \mathbb{R}^n$ with normalized measure $\mu_1(\mathcal{A}) \geq 1/2$. When using the geodesic metric, the $\epsilon$-expansion $\mathcal{A}(\epsilon)$ is at least as large as the $\epsilon$-expansion of a half sphere.*

The classical isoperimetric inequality is a simple geometric statement, and frequently appears without absolute bounds on the size of the $\epsilon$-expansion of a half-sphere, or with bounds that involve unspecified constants (Vershynin, 2017). A tight bound derived by Milman & Schechtman (1986) is given below. The asymptotic blow-up of the $\epsilon$-expansion of a half sphere predicted by this bound is shown in Figure 3.

**Lemma 2** ($\epsilon$-expansion of half sphere). *The geodesic $\epsilon$-expansion of a half sphere has normalized measure at least*

$$1 - \left(\frac{\pi}{8}\right)^{\frac{1}{2}} \exp\left(-\frac{n-1}{2}\epsilon^2\right).$$

Lemmas 1 and 2 together can be taken to mean that, if a set is not too small, then in high dimensions almost all points on the sphere are reachable within a short $\epsilon$ jump from that set. These lemmas have immediate implications for adversarial examples, which are formed by mapping one class into another using small

perturbations. Despite its complex appearance, the result below is a consequence of the (relatively simple) isoperimetric inequality.

**Theorem 1** (Existence of Adversarial Examples). *Consider a classification problem with $m$ object classes, each distributed over the unit sphere $\mathbb{S}^{n-1} \subset \mathbb{R}^n$ with density functions $\{\rho_c\}_{c=1}^m$. Choose a classifier function $\mathcal{C} : \mathbb{S}^{n-1} \to \{1, 2, \ldots, m\}$ that partitions the sphere into disjoint measurable subsets. Define the following scalar constants:*

- *Let $V_c$ denote the magnitude of the supremum of $\rho_c$ relative to the uniform density. This can be written $V_c := s_{n-1} \cdot \sup_x \rho_c(x)$.*
- *Let $f_c = \mu_1\{x | \mathcal{C}(x) = c\}$ be the fraction of the sphere labeled as $c$ by classifier $\mathcal{C}$.*

*Choose some class $c$ with $f_c \leq \frac{1}{2}$. Sample a random data point $x$ from $\rho_c$. Then with probability at least*

$$1 - V_c \left(\frac{\pi}{8}\right)^{\frac{1}{2}} \exp\left(-\frac{n-1}{2}\epsilon^2\right) \tag{1}$$

*one of the following conditions holds:*

1. *$x$ is misclassified by $\mathcal{C}$, or*
2. *$x$ admits an $\epsilon$-adversarial example in the geodesic distance.*

*Proof.* Choose a class $c$ with $f_c \leq \frac{1}{2}$. Let $\mathcal{R} = \{x | \mathcal{C}(x) = c\}$ denote the region of the sphere labeled as class $c$ by $\mathcal{C}$, and let $\overline{\mathcal{R}}$ be its complement. $\overline{\mathcal{R}}(\epsilon)$ is the $\epsilon$-expansion of $\overline{\mathcal{R}}$ in the geodesic metric. Because $\overline{\mathcal{R}}$ covers at least half the sphere, the isoperimetric inequality (Lemma 1) tells us that the epsilon expansion is at least as great as the epsilon expansion of a half sphere. We thus have

$$\mu_1[\overline{R}(\epsilon)] \geq 1 - \left(\frac{\pi}{8}\right)^{\frac{1}{2}} \exp\left(-\frac{n-1}{2}\epsilon^2\right).$$

Now, consider the set $\mathcal{S}_c$ of "safe" points from class $c$ that are correctly classified and do not admit adversarial perturbations. A point is correctly classified only if it lies inside $\mathcal{R}$, and therefore outside of $\overline{\mathcal{R}}$. To be safe from adversarial perturbations, a point cannot lie within $\epsilon$ distance from the class boundary, and so it cannot lie within $\overline{R}(\epsilon)$. It is clear that the set $\mathcal{S}_c$ of safe points is exactly the complement of $\overline{R}(\epsilon)$. This set has normalized measure

$$\mu_1[\mathcal{S}_c] \leq \left(\frac{\pi}{8}\right)^{\frac{1}{2}} \exp\left(-\frac{n-1}{2}\epsilon^2\right).$$

The probability of a random point lying in $\mathcal{S}_c$ is bounded above by the normalized supremum of $\rho_c$ times the normalized measure $\mu_1[\mathcal{S}_c]$. This product is given by

$$V_c \left(\frac{\pi}{8}\right)^{\frac{1}{2}} \exp\left(-\frac{n-1}{2}\epsilon^2\right).$$

We then subtract this probability from 1 to obtain the probability of a point lying outside the safe region, and arrive at equation 1. $\qquad\square$

In the above result, we measure the size of adversarial perturbations using the geodesic distance. Most studies of adversarial examples measure the size of perturbation in either the $\ell_2$ (Euclidean) norm or the $\ell_\infty$ (max) norm, and so it is natural to wonder whether Theorem 1 depends strongly on the distance metric. Fortunately (or, rather unfortunately) it does not.

It is easily observed that, for any two points $x$ and $y$ on a sphere,

$$d_\infty(x, y) \leq d_2(x, y) \leq d_g(x, y),$$

where $d_\infty(x, y)$, $d_2(x, y)$, and $d_g(x, y)$ denote the $l_\infty$, Euclidean, and geodesic distance, respectively. From this, we see that Theorem 1 is actually fairly conservative; any $\epsilon$-adversarial example in the geodesic metric would also be adversarial in the other two metrics, and the bound in Theorem 1 holds regardless of which of the three metrics we choose (although different values of $\epsilon$ will be appropriate depending on the norm).

## 4    WHAT ABOUT THE UNIT CUBE?

The above result about the sphere is simple and easy to prove using classical results. However, real world images do not lie on the sphere. In a more typical situation, images will be scaled so that their pixels lie in $[0, 1]$, and data lies inside a high-dimensional hypercube (but, unlike the sphere, data is not confined to its surface). The proof of Theorem 1 makes extensive use of properties that are exclusive to the sphere, and is not applicable to this more realistic setting. Are there still problem classes on the cube where adversarial examples are inevitable?

This question is complicated by the fact that *geometric* isoperimetric inequalities do not exist for the cube, as the shapes that achieve minimal $\epsilon$-expansion (if they exist) depend on the volume they enclose and the choice of $\epsilon$ (Ros, 2001). Fortunately, researchers have been able to derive "algebraic" isoperimetric inequalities that provide lower bounds on the size of the $\epsilon$-expansion of sets without identifying the shape that achieves this minimum (Talagrand, 1996; Milman & Schechtman, 1986). The result below about the unit cube is analogous to Proposition 2.8 in Ledoux (2001), except with tighter constants. For completeness, a proof (which utilizes methods from Ledoux) is provided in Appendix A.

**Lemma 3** (Isoperimetric inequality on a cube). *Consider a measurable subset of the cube $\mathcal{A} \subset [0,1]^n$, and a p-norm distance metric $d_p(x, y) = \|x - y\|_p$ for $p > 0$. Let $\Phi(z) = (2\pi)^{-\frac{1}{2}} \int_{-\infty}^{z} e^{-t^2/2} dt$, and let $\alpha$ be the scalar that satisfies $\Phi(\alpha) = \text{vol}[\mathcal{A}]$. Then*

$$\text{vol}[\mathcal{A}(\epsilon, d_p)] \geq \Phi\left(\alpha + \frac{\sqrt{2\pi n}}{n^{1/p^*}}\epsilon\right) \tag{2}$$

*where $p^* = \min(p, 2)$. In particular, if $\text{vol}(\mathcal{A}) \geq 1/2$, then we simply have*

$$\text{vol}[\mathcal{A}(\epsilon, d_p)] \geq 1 - \frac{\exp(-\pi n^{1-2/p^*}\epsilon^2)}{2\pi n^{1/2-1/p^*}}. \tag{3}$$

Using this result, we can show that most data samples in a cube admit adversarial examples, provided the data distribution is not excessively concentrated.

**Theorem 2** (Adversarial examples on the cube). *Consider a classification problem with $m$ classes, each distributed over the unit hypercube $[0,1]^n$ with density functions $\{\rho_c\}_{c=1}^m$. Choose a classifier function $\mathcal{C} : [0,1]^n \rightarrow \{1, 2, \ldots, m\}$ that partitions the hypercube into disjoint measurable subsets. Define the following scalar constants:*

- *Let $U_c$ denote the supremum of $\rho_c$.*
- *Let $f_c$ be the fraction of hypercube partitioned into class $c$ by $\mathcal{C}$.*

*Choose some class $c$ with $f_c \leq \frac{1}{2}$, and select an $\ell_p$-norm with $p > 0$. Define $p^* = \min(p, 2)$. Sample a random data point $x$ from the class distribution $\rho_c$. Then with probability at least*

$$1 - U_c \frac{\exp(-\pi n^{1-2/p^*}\epsilon^2)}{2\pi n^{1/2-1/p^*}}. \tag{4}$$

*one of the following conditions holds:*

1. *$x$ is misclassified by $\mathcal{C}$, or*
2. *$x$ has an adversarial example $\hat{x}$, with $\|x - \hat{x}\|_p \leq \epsilon$.*

When adversarial examples are defined in the $\ell_2$-norm (or for any $p \geq 2$), the bound in equation 4 becomes

$$1 - U_c \exp(-\pi\epsilon^2)/(2\pi). \tag{5}$$

Provided the class distribution is not overly concentrated, equation 5 guarantees adversarial examples with relatively "small" $\epsilon$ relative to a typical vector. In $n$ dimensions, the $\ell_2$ diameter of the cube is $\sqrt{n}$, and so it is reasonable to choose $\epsilon = O(\sqrt{n})$ in equation 5. In Figure 1, we chose $\epsilon = 10$. A similarly strong bound of $1 - U_c\sqrt{n}\exp(-\pi\epsilon^2/n)/(2\pi)$ holds for the case of the $\ell_1$-norm, in which case the diameter is $n$.

Oddly, equation 4 seems particularly weak when the $\ell_\infty$ norm is used. In this case, the bound on the right side of equation 4 becomes equation 5 just like in the $\ell_2$ case. However, $\ell_\infty$ adversarial examples are only interesting if we take $\epsilon < 1$, in which case equation 5 becomes vacuous for large $n$. This bound can be tightened up in certain situations. If we prove Theorem 2 using the tighter (but messier) bound of equation 2 instead of equation 3, we can replace equation 4 with

$$1 - U_c \hat{\Phi}\left(\alpha + \sqrt{2\pi}\epsilon\right)$$

for $p \geq 2$, where $\hat{\Phi}(z) = \frac{1}{\sqrt{2\pi}}\int_z^\infty e^{-t^2/2}dt \geq \frac{1}{\sqrt{2\pi}z}e^{-z^2/2}$ (for $z > 0$), and $\alpha = \Phi^{-1}(1 - f_c)$. For this bound to be meaningful with $\epsilon < 1$, we need $f_c$ to be relatively small, and $\epsilon$ to be roughly $f_c$ or smaller. This is realistic for some problems; ImageNet has 1000 classes, and so $f_c < 10^{-3}$ for at least one class.

Interestingly, under $\ell_\infty$-norm attacks, guarantees of adversarial examples are much stronger on the sphere (Section 3) than on the cube. One might wonder whether the weakness of Theorem 4 in the $\ell_\infty$ case is fundamental, or if this is a failure of our approach. One can construct examples of sets with $\ell_\infty$ expansions that nearly match the behavior of equation 5, and so our theorems in this case are actually quite tight. It seems to be inherently more difficult to prove the existence of adversarial examples in the cube using the $\ell_\infty$-norm.

## 5    WHAT ABOUT SPARSE ADVERSARIAL EXAMPLES?

A number of papers have looked at *sparse* adversarial examples, in which a small number of image pixels, in some cases only one (Su et al., 2017), are changed to manipulate the class label. To study this case, we would like to investigate adversarial examples under the $\ell_0$ metric. The $\ell_0$ distance is defined as

$$d(x, y) = \|x - y\|_0 = \text{card}\{i \mid x_i \neq y_i\}.$$

If a point $x$ has an $\epsilon$-adversarial example in this norm, then it can be perturbed into a different class by modifying at most $\epsilon$ pixels (in this case $\epsilon$ is taken to be a positive integer).

Theorem 2 is fairly tight for $p = 1$ or $2$. However, the bound becomes quite loose for small $p$, and in particular it fails completely for the important case of $p = 0$. For this reason, we present a different bound that is considerably tighter for small $p$ (although slightly looser for large $p$).

The case $p = 0$ was studied by Milman & Schechtman (1986) (Section 6.2) and McDiarmid (1989), and later by Talagrand (1995; 1996). The proof of the following theorem (appendix B) follows the method used in Section 5 of Talagrand (1996), with modifications made to extend the proof to arbitrary $p$.

**Lemma 4** (Isoperimetric inequality on the cube: small $p$). *Consider a measurable subset of the cube $\mathcal{A} \subset [0, 1]^n$, and a p-norm distance metric $d(x, y) = \|x - y\|_p$ for any $p \geq 0$. We have*

$$\text{vol}[\mathcal{A}(\epsilon, d_p)] \geq 1 - \frac{\exp\left(-\epsilon^{2p}/n\right)}{\text{vol}[\mathcal{A}]}, \text{ for } p > 0 \text{ and} \tag{6}$$

$$\text{vol}[\mathcal{A}(\epsilon, d_0)] \geq 1 - \frac{\exp\left(-\epsilon^2/n\right)}{\text{vol}[\mathcal{A}]}, \text{ for } p = 0. \tag{7}$$

Using this result, we can prove a statement analogous to Theorem 2, but for sparse adversarial examples. We present only the case of $p = 0$, but the generalization to the case of other small $p$ using Lemma 4 is straightforward.

**Theorem 3** (Sparse adversarial examples). *Consider the problem setup of Theorem 2. Choose some class $c$ with $f_c \leq \frac{1}{2}$, and sample a random data point $x$ from the class distribution $\rho_c$. Then with probability at least*

$$1 - 2U_c \exp(-\epsilon^2/n) \tag{8}$$

*one of the following conditions holds:*

1. *$x$ is misclassified by $\mathcal{C}$, or*

2. *$x$ can be adversarially perturbed by modifying at most $\epsilon$ pixels, while still remaining in the unit hypercube.*

## 6 WHAT IF WE JUST SHOW THAT ADVERSARIAL EXAMPLES EXIST?

Tighter bounds can be obtained if we only guarantee that adversarial examples exist for *some* data points in a class, without bounding the probability of this event.

**Theorem 4** (Condition for existence of adversarial examples). *Consider the setup of Theorem 2. Choose a class $c$ that occupies a fraction of the cube $f_c < \frac{1}{2}$. Pick an $\ell_p$ norm and set $p^* = \min(p, 2)$.*

*Let* $\operatorname{supp}(\rho_c)$ *denote the support of* $\rho_c$. *Then there is a point $x$ with $\rho_c(x) > 0$ that admits an $\epsilon$-adversarial example if*

$$\operatorname{vol}[\operatorname{supp}(\rho_c)] \geq \begin{cases} \frac{1}{2}\exp(-\pi\epsilon^2 n^{1-2/p^*}), & \text{for } p > 0 \text{ or} \\ \exp\left(-2\left(\epsilon - \sqrt{\frac{n\log 2}{2}}\right)^2 / n\right), & \text{for } p = 0. \end{cases} \tag{9}$$

*The bound for the case $p = 0$ is valid only if $\epsilon \geq \sqrt{n\log 2/2}$.*

It is interesting to consider when Theorem 4 produces non-vacuous bounds. When the $\ell_2$-norm is used, the bound becomes $\operatorname{vol}[\operatorname{supp}(\rho_c)] \geq \exp(-\pi\epsilon^2)/2$. The diameter of the cube is $\sqrt{n}$, and so the bound becomes active for $\epsilon = \sqrt{n}$. Plugging this in, we see that the bound is active whenever the size of the support satisfies $\operatorname{vol}[\operatorname{supp}(\rho_c)] > \frac{1}{2e^{\pi n}}$. Remarkably, this holds for large $n$ whenever the support of class $c$ is larger than (or contains) a hypercube of side length at least $e^{-\pi} \approx 0.043$. Note, however, that the bound being "active" does not guarantee adversarial examples with a "small" $\epsilon$.

## 7 DISCUSSION: CAN WE ESCAPE FUNDAMENTAL BOUNDS?

There are a number of ways to escape the guarantees of adversarial examples made by Theorems 1-4. One potential escape is for the class density functions to take on extremely large values (i.e., exponentially large $U_c$); the dependence of $U_c$ on $n$ is addressed separately in Section 8.

**Unbounded density functions and low-dimensional data manifolds**   In practice, image datasets might lie on low-dimensional manifolds within the cube, and the support of these distributions could have measure zero, making the density function infinite (i.e., $U_c = \infty$). The arguments above are still relevant (at least in theory) in this case; we can expand the data manifold by adding a uniform random noise to each image pixel of magnitude at most $\epsilon_1$. The expanded dataset has positive volume. Then, adversarial examples of this expanded dataset can be crafted with perturbations of size $\epsilon_2$. This method of expanding the manifold before crafting adversarial examples is often used in practice. Tramèr et al. (2017a) proposed adding a small perturbation to step off the image manifold before crafting adversarial examples. This strategy is also used during adversarial training (Madry et al., 2017).

**Adding a "don't know" class**   The analysis above assumes the classifier assigns a label to every point in the cube. If a classifier has the ability to say "I don't know," rather than assign a label to every input, then the region of the cube that is assigned class labels might be very small, and adversarial examples could be escaped even if the other assumptions of Theorem 4 are satisfied. In this case, it would still be easy for the adversary to degrade classifier performance by perturbing images into the "don't know" class.

**Feature squeezing**   If decreasing the dimensionality of data does not lead to substantially increased values for $U_c$ (we see in Section 8 that this is a reasonable assumption) or loss in accuracy (a stronger assumption), measuring data in lower dimensions could increase robustness. This can be done via an auto-encoder (Meng & Chen, 2017; Shen et al., 2017), JPEG encoding (Das et al., 2018), or quantization (Xu et al., 2017).

**Computational hardness**   It may be computationally hard to craft adversarial examples because of local flatness of the classification function, obscurity of the classifier function, or other computational difficulties. Computational hardness could prevent adversarial attacks in practice, even if adversarial examples still exist.

## 8 EXPERIMENTS & EFFECT OF DIMENSIONALITY

In this section, we discuss the relationship between dimensionality and adversarial robustness, and explore how the predictions made by the theorems above are reflected in experiments.

It is commonly thought that high-dimensional classifiers are more susceptible to adversarial examples than low-dimensional classifiers. This perception is partially motivated by the observation that classifiers on high-resolution image distributions like ImageNet are more easily fooled than low resolution classifiers on MNIST (Tramèr et al., 2017a). Indeed, Theorem 2 predicts that high-dimensional classifiers should be much easier to fool than low-dimensional classifiers, assuming the datasets they classify have comparable probability density limits $U_c$. However, this is not a reasonable assumption; we will see below that high dimensional distributions may be more concentrated than their low-dimensional counterparts.

We study the effects of dimensionality with a thought experiment involving a "big MNIST" image distribution. Given an integer expansion factor $b$, we can make a big MNIST distribution, denoted $b$-MNIST, by replacing each pixel in an MNIST image with a $b \times b$ array of identical pixels. This expands an original $28 \times 28$ image into a $28b \times 28b$ image. Figure 4a shows that, without adversarial training, a classifier on big MNIST is far more susceptible to attacks than a classifier trained on the original MNIST[1].

However, each curve in Figure 4a only shows the attack susceptibility of one particular classifier. In contrast, Theorems 1-4 describe the *fundamental* limits of susceptibility for all classifiers. These limits are an inherent property of the data distribution. The theorem below shows that these fundamental limits *do not* depend in a non-trivial way on the dimensionality of the images in big MNIST, and so the relationship between dimensionality and susceptibility in Figure 4a results from the weakness of the training process.

**Theorem 5.** *Suppose $\epsilon$ and $p$ are such that, for all MNIST classifiers, a random image from class $c$ has an $\epsilon$-adversarial example (in the $\ell_2$-norm) with probability at least $p$. Then for all classifiers on $b$-MNIST, with integer $b \geq 1$, a random image from $c$ has a $b\epsilon$-adversarial example with probability at least $p$.*

*Likewise, if all $b$-MNIST classifiers have $b\epsilon$-adversarial examples with probability $p$ for some $b \geq 1$, then all classifiers on the original MNIST distribution have $\epsilon$-adversarial examples with probability $p$.*

Theorem 5 predicts that the perturbation needed to fool all $56 \times 56$ classifiers is twice that needed to fool all $28 \times 28$ classifiers. This is reasonable since the $\ell_2$-norm of a $56 \times 56$ image is twice that of its $28 \times 28$ counterpart. Put simply, fooling big MNIST is just as hard/easy as fooling the original MNIST regardless of resolution. This also shows that for big MNIST, as the expansion factor $b$ gets larger and $\epsilon$ is expanded to match, the concentration bound $U_c$ grows at exactly the same rate as the exponential term in equation 2 shrinks, and there is no net effect on fundamental susceptibility. Also note that an analogous result could be based on any image classification problem (we chose MNIST only for illustration), and any $p \geq 0$.

We get a better picture of the fundamental limits of MNIST by considering classifiers that are hardened by adversarial training[2] (Figure 4b). These curves display several properties of fundamental limits predicted by our theorems. As predicted by Theorem 5, the $112 \times 112$ classifer curve is twice as wide as the $56 \times 56$ curve, which in turn is twice as wide as the $28 \times 28$ curve. In addition, we see the kind of "phase transition" behavior predicted by Theorem 2, in which the classifier suddenly changes from being highly robust to being highly susceptible as $\epsilon$ passes a critical threshold. For these reasons, it is reasonable to suspect that the adversarially trained classifiers in Figure 4b are operating near the fundamental limit predicted by Theorem 2.

Theorem 5 shows that increased dimensionality does not increase adversarial susceptibility in a fundamental way. But then why are high-dimensional classifiers so easy to fool? To answer this question, we look at the concentration bound $U_c$ for object classes. The smallest possible value of $U_c$ is 1, which only occurs when images are "spread out" with uniform, uncorrelated pixels. In contrast, adjacent pixels in MNIST (and especially big MNIST) are very highly correlated, and images are concentrated near simple, low-dimensional manifolds, resulting in highly concentrated image classes with large $U_c$. Theory predicts that such highly concentrated datasets can be relatively safe from adversarial examples.

---

[1] Only the fully-connected layer is modified to handle the difference in dimensionality between datasets.

[2] Adversarial examples for MNIST/CIFAR-10 were produced as in Madry et al. (2017) using 100-step/20-step PGD.

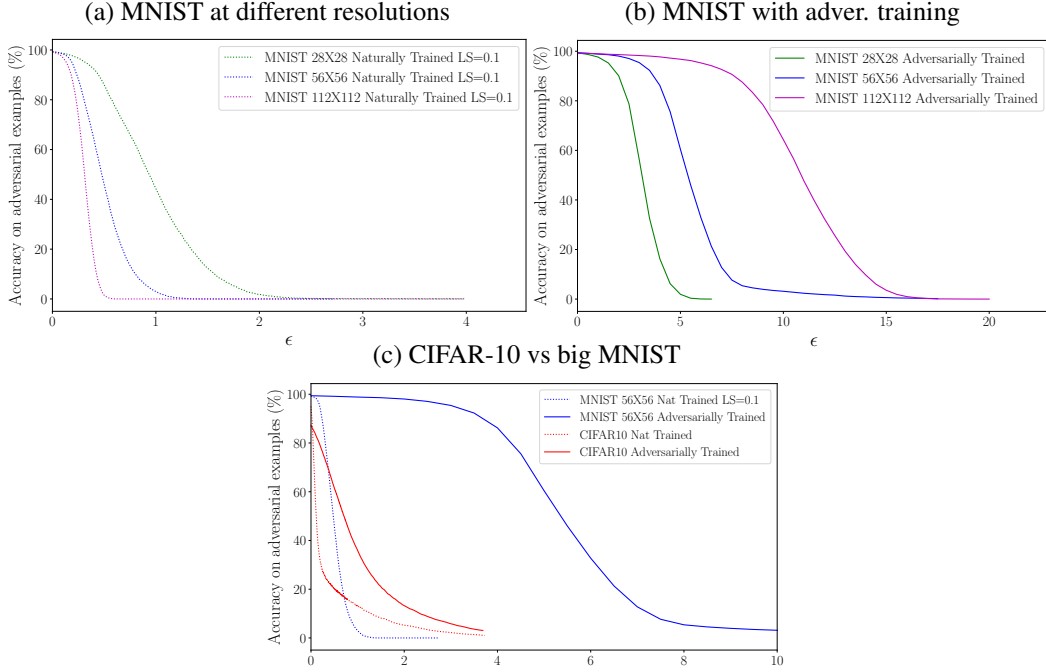

Figure 4: (a) Robustness of MNIST and "big" MNIST classifiers as a function of $\epsilon$. Naturally trained classifiers are less robust with increased dimensionality. (b) With adversarial training, susceptibility curves behave as predicted by Theorems 2 and 5. (c) The susceptibility of CIFAR-10 is compared to big MNIST. Both datasets have similar dimension, but the higher complexity of CIFAR-10 results in far worse susceptibility. Perturbations are measured in the $\ell_2$-norm.

We can reduce $U_c$ and dramatically increase susceptibility by choosing a more "spread out" dataset, like CIFAR-10, in which adjacent pixels are less strongly correlated and images appear to concentrate near complex, higher-dimensional manifolds. We observe the effect of decreasing $U_c$ by plotting the susceptibility of a $56 \times 56$ MNIST classifier against a classifier for CIFAR-10 (Figure 4, right). The former problem lives in 3136 dimensions, while the latter lives in 3072, and both have 10 classes. Despite the structural similarities between these problems, the decreased concentration of CIFAR-10 results in vastly more susceptibility to attacks, regardless of whether adversarial training is used. The theory above suggests that this increased susceptibility is caused at least in part by a shift in the fundamental limits for CIFAR-10, rather than the weakness of the particular classifiers we chose.

Informally, the concentration limit $U_c$ can be interpreted as a measure of image *complexity*. Image classes with smaller $U_c$ are likely concentrated near high-dimensional complex manifolds, have more intra-class variation, and thus more apparent complexity. An informal interpretation of Theorem 2 is that "high complexity" image classes are fundamentally more susceptible to adversarial examples, and Figure 4 suggests that complexity (rather than dimensionality) is largely responsible for differences we observe in the effectiveness of adversarial training for different datasets.

## 9 SO...ARE ADVERSARIAL EXAMPLES INEVITABLE?

The question of whether adversarial examples are inevitable is an ill-posed one. Clearly, any classification problem has a fundamental limit on robustness to adversarial attacks that cannot be escaped by any classifier. However, we have seen that these limits depend not only on fundamental properties of the dataset, but also on the strength of the adversary and the metric used to measure perturbations. This paper provides a characterization of these limits and how they depend on properties of the data distribution. Unfortunately, it is impossible to know the exact properties of real-world image distributions or the resulting fundamental limits of adversarial training for specific datasets. However, the analysis and experiments in this paper suggest that, especially for complex image classes in high-dimensional spaces, these limits may be far worse than our intuition tells us.

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

## A  PROOF OF LEMMA 3

We now prove Lemma 3. To do this, we begin with a classical isoperimetric inequality for random Gaussian variables. Unlike the case of a cube, tight geometric isoperimetric inequalities exist in this case. We then prove results about the cube by creating a mapping between uniform random variables on the cube and random Gaussian vectors.

In the lemma below, we consider the standard Gaussian density in $\mathbb{R}^n$ given by $p(x) = \frac{1}{(2\pi)^{n/2}} e^{-nx^2/2}$ and corresponding Gaussian measure $\mu$. We also define

$$\Phi(z) = \frac{1}{\sqrt{2\pi}} \int_{-\infty}^{z} e^{-t^2/2} dt,$$

which is the cumulative density of a Gaussian curve.

The following Lemma was first proved in Sudakov & Tsirelson (1974), and an elementary proof was given in Bobkov et al. (1997).

**Lemma 5** (Gaussian Isoperimetric Inequality). *Of all sets with the same Gaussian measure, the set with $\ell_2$ $\epsilon$-expansion of smallest measure is a half space. Furthermore, for any measurable set $\mathcal{A} \subset \mathbb{R}^n$, and scalar constant $a$ such that $\Phi(a) = \mu[\mathcal{A}]$,*

$$\mu[\mathcal{A}(\epsilon, d_2)] \geq \Phi(a + \epsilon).$$

Using this result we can now give a proof of Lemma 3.

This function $\Phi$ maps a random *Guassian* vector $z \in N(0, I)$ onto a random *uniform* vector in the unit cube. To see why, consider a measurable subset $\mathcal{B} \subset R^n$. If $\mu$ is the Gaussian measure on $\mathbb{R}^n$ and $\sigma$ is the uniform measure on the cube, then

$$\sigma[\Phi(\mathcal{B})] = \int \chi_{\Phi(\mathcal{B})}(s) d\sigma = \int \chi_{\Phi(\mathcal{B})}(\Phi(z)) \frac{1}{\det(J\Phi)} dz = \int \chi_{\mathcal{B}}(z) d\mu = \mu[\mathcal{B}].$$

Since $\frac{\partial}{\partial z_i} \Phi(z) \leq \frac{1}{\sqrt{2\pi}}$, we also have

$$\|\Phi(z) - \Phi(w)\|_p \leq \|\frac{1}{\sqrt{2\pi}}(x - w)\|_p = \frac{1}{\sqrt{2\pi}} \|z - w\|_p$$

for any $z, w \in \mathbb{R}^n$. From this, we see that for $p^* = \min(p, 2)$

$$\|\Phi(z) - \Phi(w)\|_p \leq n^{1/p^* - 1/2}\|\Phi(z) - \Phi(w)\|_2 \leq \frac{n^{1/p^*}}{\sqrt{2\pi n}}\|z - w\|_2 \tag{10}$$

where we have used the identity $\|u\|_p \leq n^{1/\min(p,2) - 1/2}\|u\|_2$.

Now, consider any set $\mathcal{A}$ in the cube, and let $\mathcal{B} = \Phi^{-1}(\mathcal{A})$. From equation 10, we see that

$$\Phi\mathcal{B}\left(\frac{\sqrt{2\pi n}}{n^{1/p^*}}\epsilon, d_2\right) \subset \mathcal{A}(\epsilon, d_p).$$

It follows from equation 10 that

$$\sigma[\mathcal{A}(\epsilon, d_p)] \geq \mu\left[\mathcal{B}\left(\frac{\sqrt{2\pi n}}{n^{1/p^*}}\epsilon, d_2\right)\right].$$

Applying Lemma 5, we see that

$$\sigma[\mathcal{A}(\epsilon, d_p)] \geq \Phi\left(\alpha + \frac{\sqrt{2\pi n}}{n^{1/p^*}}\epsilon\right) \tag{11}$$

where $\alpha = \Phi^{-1}(\sigma[\mathcal{A}])$.

To obtain the simplified formula in the theorem, we use the identity

$$\frac{1}{\sqrt{2\pi}}\int_x^\infty e^{-t^2}dt < \frac{e^{-x^2}}{\sqrt{2\pi}x}$$

which is valid for $x > 0$, and can be found in Abramowitz & Stegun (1965).

## B  PROOF OF LEMMA 4

Our proof emulates the method of Talagrand, with minor modifications that extend the result to other $\ell_p$ norms. We need the following standard inequality. Proof can be found in Talagrand (1995; 1996).

**Lemma 6** (Talagrand). *Consider a probability space $\Omega$ with measure $\mu$. For $g : \Omega \to [0, 1]$, we have*

$$\int_\Omega \min\left(e^t, \frac{1}{g^\alpha}\right)d\mu \times \left(\int_\Omega g d\mu\right)^\alpha \leq \exp\left(\frac{t^2(\alpha + 1)}{8\alpha}\right).$$

Our proof of Lemma 3 follows the three-step process of Talagrand illustrated in Talagrand (1995). We begin by proving the bound

$$\int e^{tf(x,\mathcal{A})}dx \leq \frac{1}{\sigma^\alpha[\mathcal{A}]}\exp\left(\frac{nt^2(\alpha + 1)}{8\alpha}\right) \tag{12}$$

where $f(x, \mathcal{A}) = \min_{y \in \mathcal{A}}\sum_i |x_i - y_i|^p = d_p^p(x, \mathcal{A})$ is a measure of distance from $\mathcal{A}$ to $x$, and $\alpha, t$ are arbitrary positive constants. Once this bound is established, a Markov bound can be used to obtain the final result. Finally, constants are tuned in order to optimize the tightness of the bound.

We start by proving the bound in equation 12 using induction on the dimension. The base case for the induction is $n = 1$, and we have

$$\int e^{tf(x,\mathcal{A})}dx \leq \sigma[\mathcal{A}] + \int_{\mathcal{A}^c} e^{tf(x,\mathcal{A})}dx \leq \sigma[\mathcal{A}] + \int_{\mathcal{A}^c} 1 dx \leq \sigma[\mathcal{A}] + (1 - \sigma[\mathcal{A}])e^t \leq \frac{1}{\sigma^\alpha[\mathcal{A}]}\exp\left(\frac{t^2(\alpha + 1)}{8\alpha}\right).$$

We now prove the result for $n$ dimensions using the inductive hypothesis. We can upper bound the integral by integrating over "slices" along one dimension. Let $\mathcal{A} \subset [0, 1]^n$. Define

$$\mathcal{A}_\omega = \{z \in \mathbb{R}^{n-1} \,|\, (\omega, z) \in \mathcal{A}\} \text{ and } \mathcal{B} = \{z \in \mathbb{R}^{n-1} \,|\, (\omega, z) \in \mathcal{A} \text{ for some } \omega\}.$$

Clearly, the distance from $(\omega, z)$ to $\mathcal{A}$ is at most the distance from $z$ to $\mathcal{A}_\omega$, and so

$$\int e^{tf(x,\mathcal{A})}dx \leq \int_{\omega \in [0,1]} \int_{z \in [0,1]^{n-1}} e^{tf(z,\mathcal{A}_\omega)}dz\, dx \leq \int_{\omega \in [0,1]} \frac{1}{\sigma^\alpha[\mathcal{A}_\omega]} \exp\left(\frac{(n-1)t^2(\alpha+1)}{8\alpha}\right).$$

We also have that the distance from $x$ to $\mathcal{A}$ is at most one unit greater than the distance from $x$ to $\mathcal{B}$. This gives us

$$\int e^{tf(x,\mathcal{A})}dx \leq \int_{(\omega,z)\in[0,1]^n} e^{t(f(x,\mathcal{B})+1)} \leq e^t \int_{(\omega,z)\in[0,1]^n} e^{tf(x,\mathcal{B})} \leq \frac{e^t}{\sigma^\alpha[\mathcal{B}]} \exp\left(\frac{(n-1)t^2(\alpha+1)}{8\alpha}\right).$$

Applying equation 6 gives us

$$\int e^{tf(x,\mathcal{A})}dx \leq \int_{\omega\in[0,1]} \min\left(\frac{e^t}{\sigma^\alpha[\mathcal{B}]} \exp\left(\frac{(n-1)t^2(\alpha+1)}{8\alpha}\right), \frac{1}{\sigma^\alpha[\mathcal{A}_\omega]} \exp\left(\frac{(n-1)t^2(\alpha+1)}{8\alpha}\right)\right)$$
$$= \exp\left(\frac{(n-1)t^2(\alpha+1)}{8\alpha}\right) \frac{1}{\sigma^\alpha[\mathcal{B}]} \int_{\omega\in[0,1]} \min\left(e^t, \frac{\sigma^\alpha[\mathcal{B}]}{\sigma^\alpha[\mathcal{A}_\omega]}\right). \tag{13}$$

Now, we apply lemma 6 to equation 13 with $g(\omega) = \alpha[\mathcal{A}_\omega]/\alpha[\mathcal{B}]$ to arrive at equation 12.

The second step of the proof is to produce a Markov inequality from equation 12. For the bound in equation 12 to hold, we need

$$1 - \sigma[\mathcal{A}(\epsilon, d_p)] = \sigma\{x \mid f(x) > \epsilon^p\} \leq \frac{\int e^{tf(x,A)}dx}{e^{t\epsilon^p}} \leq \frac{\exp\left(\frac{nt^2(\alpha+1)}{8\alpha}\right)}{\sigma^\alpha[\mathcal{A}]e^{t\epsilon^p}}. \tag{14}$$

The third step is to optimize the bound by choosing constants. We minimize the right hand side by choosing $t = \frac{4\alpha\epsilon^p}{n(\alpha+1)}$ to get

$$1 - \sigma[\mathcal{A}(\epsilon, d_p)] \leq \frac{\exp\left(-\frac{2\alpha\epsilon^{2p}}{n(\alpha+1)}\right)}{\sigma^\alpha[\mathcal{A}]}. \tag{15}$$

Now, we can simply choose $\alpha = 1$ to get the simple bound

$$1 - \sigma[\mathcal{A}(\epsilon, d_p)] \leq \frac{\exp\left(-\epsilon^{2p}/n\right)}{\sigma[\mathcal{A}]}, \tag{16}$$

or we can choose the optimal value of $\alpha = \sqrt{\frac{2\epsilon^{2p}}{n\log(1/\sigma)}} - 1$, which optimizes the bound in the case $\epsilon^{2p} \geq \frac{n}{2}\log(1/\sigma(\mathcal{A}))$. We arrive at

$$1 - \sigma[\mathcal{A}(\epsilon, d_p)] \leq \exp\left(-\frac{2}{n}\left(\epsilon^p - \sqrt{n\log(\sigma^{-1}[\mathcal{A}])/2}\right)^2\right). \tag{17}$$

This latter bound is stronger than we need to prove Lemma 3, but it will come in handy later to prove Theorem 4.

## C   PROOF OF THEOREMS 2 AND 3

We combine the proofs of these results since their proofs are nearly identical. The proofs closely follow the argument of Theorem 1.

Choose a class $c$ with $f_c \leq \frac{1}{2}$ and let $\mathcal{R} = \{x | \mathcal{C}(x) = c\}$ denote the subset of the cube lying in class $c$ according to the classifier $\mathcal{C}$. Let $\overline{\mathcal{R}}$ be the complement, who's $\ell_p$ expansion is denoted $\overline{\mathcal{R}}(\epsilon; d_p)$. Because $\overline{\mathcal{R}}$ covers at least half the cube, we can invoke Lemma 3. We have that

$$\text{vol}[\overline{\mathcal{R}}(\epsilon; h)] \geq 1 - \delta,$$

where

$$\delta = \begin{cases} \frac{\exp(-\pi n^{1-2/p^*}\epsilon^2)}{2\pi n^{1/2-1/p^*}}, & \text{for Theorem 2 and} \\ 2U_c \exp(-\epsilon^2/n), & \text{for Theorem 3.} \end{cases} \tag{18}$$

The set $\overline{\overline{\mathcal{R}(\epsilon; h)}}$ contains all points that are correctly classified and safe from adversarial perturbations. This region has volume at most $\delta$, and the probability of a sample from the class distribution $\rho_c$ lying in this region is at most $U_c\delta$. We then subtract this from 1 to obtain the mass of the class distribution lying in the "unsafe" region $\overline{\mathcal{R}}_c$.

## D  PROOF OF THEOREM 4

Let $\mathcal{A}$ denote the support of $p_c$, and suppose that this support has measure $\mathrm{vol}[\mathcal{A}] = \eta$. We want to show that, for large enough $\epsilon$, the expansion $\mathcal{A}(\epsilon, d_p)$ is larger than half the cube. Since class $c$ occupies less than half the cube, this would imply that $\mathcal{A}(\epsilon, d_p)$ overlaps with other classes, and so there must be data points in $\mathcal{A}$ with $\epsilon$-adversarial examples.

We start with the case $p > 0$, where we bound $\mathcal{A}(\epsilon, d_p)$ using equation 2 of Lemma 3. To do this, we need to approximate $\Phi^{-1}(\eta)$. This can be done using the inequality

$$\Phi(\alpha) = \frac{1}{2\pi} \int_{-\infty}^{\alpha} e^{-t^2/2} dt \le \frac{1}{2} e^{-\alpha^2/2},$$

which holds for $\alpha < 0$. Rearranging, we obtain

$$\alpha \ge -\sqrt{\log \frac{1}{4\Phi(\alpha)^2}}. \tag{19}$$

Now, if $\alpha = \Phi^{-1}(\eta)$, then $\Phi(\alpha) = \eta$, and equation 19 gives us $\alpha \ge -\sqrt{\log \frac{1}{4\eta^2}}$. Plugging this into equation 2 of Lemma 3, we get

$$\mathrm{vol}[\mathcal{A}(\epsilon, d_p)] \ge \Phi(\alpha + \epsilon) \ge \Phi\left(-\sqrt{\log \frac{1}{4\eta^2}} + \frac{\sqrt{2\pi n}}{n^{1/p^*}}\epsilon\right).$$

The quantity on the left will be greater than $\frac{1}{2}$, thus guaranteeing adversarial examples, if

$$\frac{\sqrt{2\pi n}}{n^{1/p^*}}\epsilon > \sqrt{\log \frac{1}{4\eta^2}}.$$

This can be re-arranged to obtain the desired result.

In the case $p = 0$, we need to use equation 17 from the proof of Lemma 3 in Appendix B, which we restate here

$$\mathrm{vol}[\mathcal{A}(\epsilon, d_0)] \ge 1 - \exp\left(-\frac{2}{n}\left(\epsilon - \sqrt{n \log(1/\eta)/2}\right)^2\right).$$

This bound is valid, and produces a non-vacuous guarantee of adversarial examples, if

$$\exp\left(-\frac{2}{n}\left(\epsilon - \sqrt{n \log(1/\eta)/2}\right)^2\right) < \frac{1}{2}.$$

which holds if

$$\eta > \exp\left(-\frac{2\left(\epsilon - \sqrt{n \log 2/2}\right)^2}{n}\right).$$

# E   PROOF OF THEOREM 5

Assume that any MNIST classifier can be fooled by perturbations of size at most $\epsilon$ with probability at least $p$. To begin, we put a bound on the susceptibility of any $b$-MNIST classifier (for $b \geq 1$) under this assumption. We can classify MNIST images by upsampling them to resolution $28b \times 28b$ and feeding them into a high-resolution "back-end" classifier. After upsampling, an MNIST image with perturbation of norm $\epsilon$ becomes a $28b \times 28b$ image with perturbation of norm $b\epsilon$. The classifier we have constructed takes low-resolution images as inputs, and so by assumption it is fooled with probability at least $p$. However, the low-resolution classifier is fooled only when the high-resolution "back-end" classifier is fooled, and so the high-resolution classifier is fooled with probability at least $p$ at well. Note that we can build this two-scale classifier using any high-resolution classifier as a back-end, and so this bound holds uniformly over all high-resolution classifiers.

Likewise, suppose we classify $b$-MNIST images (for integer $b \geq 1$) by downsampling them to the original $28 \times 28$ resolution (by averaging pixel blocks) and feeding them into a "back-end" low-resolution classifier. After downsampling, a $28b \times 28b$ image with perturbation of norm $b\epsilon$ becomes a $28 \times 28$ image with perturbation of norm at most $\epsilon$. Whenever the high-resolution classifier is fooled, it is only because the back-end classifier is fooled by a perturbation of size at most $\epsilon$, and this happens with probability at least $p$.

