# OpenReview forum: "Are adversarial examples inevitable?"
_ICLR.cc/2019/Conference_

### Official Review · AnonReviewer1 · 2018-11-02
**a good angle, limited technical contributions, inconclusive statements**

**Rating:** 6
**Confidence:** 4

**Review:**

This paper explores the inevitability of adversarial examples with concentration inequalities. It is motivated by the difficulties of achieving adversarial robustness in literature. It derives isoperimetric inequalities on a cube, and then discuss the adversarial robustness of data distributed inside the cube, with the assumption that the data has bounded density. These inequalities are established on different norms. The authors then discuss limitation of the proposed bounds when analyzing practical data distribution and discussed the influence of dimensionality on adversarial robustness.


Novelty of the idea:
The idea of using concentration inequalities to explain vulnerability is novel in the field of adversarial examples and is a relevant/meaningful angle on understanding this phenomenon. (Although there are concurrent works also relating concentration inequalities to adversarial robustness, they don't diminish the novelty of this work.)



On technical contributions:
In summary, this paper applies / adapts previous results in concentration inequalities to develop bounds related to adversarial examples. The bounds in Lemma 3 are on any p>0, this seems to be new to my knowledge, but the technical contribution in the proof is limited.

Here are some detailed comments.

The authors claim that
"This question is complicated by the fact that simple, geometric isoperimetric inequalities fail to exist for the cube, and the shapes that achieve minimal \eps-expansion (if they exist) depend on the volume they enclose and the choice of \eps."
This statement is at least misleading, if not wrong. It is well known that geometric isoperimetric inequality does exist for cube for the L2 case (see Ledoux, M., 2001. Proposition 2.8.), and the proof procedure the author used is also very similar to the proofs in Ledoux, M., 2001.

Theorem 5's proof is confusing, if not wrong.
This is my brief recap on the first part of Thm 5,
If there exists eps and p such that, for all classifiers on MNIST, a random image has eps-adv with probability at least p, then for all classifiers on b-MNIST, a random image has b*eps-adv with probability at least p.
The proof in Appendix E says b-MNIST images can be classified by first downsampling. These downsampled classifiers do not cover "all classifiers on b-MNIST", so I don't see how the proof stands.
Likewise, the proof of the second part has the similar problem.
Therefore, I'm not yet convinced that Thm 5 is correct.
Also I suggest the authors use more rigorous language to present Theorem 5, in a similar fashion to previous theorems.

Re: Lemma 4, my understanding is that it is from previous literature. The authors should point out exactly where is it from (with section# and theorem#), so that readers and reviewers can more easily check the correctness of it.

The authors mention that "Intuitively, the concentration limit Uc can be interpreted as a measure of image complexity."
I think this statement is problematic. It is, at best, oversimplifying the the problem. If we assume the data lies in low-dimensional space, the volume of the support will be 0, no matter how complex the shape of the manifold is. This lead to unbounded density in the ambient dimension.
Even when considering "expanded dataset" like the authors discussed in Section 7, it is not obvious that Uc can be interpreted as image complexity. To make such a claim, more assumptions need to made and more analyses need to be done.
Similar comments applies to the "correlations between pixels" and concentration.



On the significance:
As the author themselves have already mentioned, the bounds described in the paper all depends on the bounded density of the data distribution. In practice, the density of data distribution is difficult to understand, if not impossible. Therefore it is still inconclusive whether the "inevitability" exists. But to be fair, I believe this is mostly due to the difficulty of the problem being studied.



Clarity and writing:
The skeleton of the paper is well written and easy to follow. I've pointed out some problems in my previous comments.
I also appreciate that the authors made efforts to not overclaim.

here are a few more comments:
- I personally feel Section 3 as an "warm-up" section is redundant, and the authors can consider move them to the appendix.
- In Section 6 and 7, the authors talk about when is the bound "meaningful" and "active". This part is confusing/misleading. eps=sqrt(n) is actually the maximum possible perturbation and not falls into the common "adversarial perturbation" where the perturbation does not change the semantic meaning of the image. There should be a least an additional numerical examples on small eps, so the readers have better ideas on the tightness/looseness of the bound.



References:
Ledoux, M., 2001. The concentration of measure phenomenon (No. 89). American Mathematical Soc..

==========================
I change my rating on this paper to be 6, after the authors' response.

---

> ### Author Response · Authors · 2018-11-25
> **Thanks for pointing out links to the isoperimetric literature.  The proof of Theorem 5 has been revised for clarity.**
>
> We thank the reviewer for considering our paper and for giving it a careful read.  We agree with the reviewer that this paper does not make any groundbreaking contributions to the field of isoperimetric inequalities.  Indeed, that was not our goal.  This is a paper on adversarial examples; we are trying to show how known results from the isoperimetry literature can be adapted to study and explain adversarial behavior in complex classifiers.  Furthermore, we made great efforts to give proper credit by citing the mathematicians who developed the isoperimetric inequalities that we rely on, and we dug into the literature to cite the original authors when possible, rather than review articles. We also took efforts to cite the authors who developed proof techniques that we use.
>
> We have revised the statement "This question is complicated by the fact..." so that it is not misinterpreted.  We want to clarify what we meant by the statement.   On the sphere, there are “geometric”  isoperimetric inequalities because we know the shapes that produce minimal epsilon-expansions (i.e., semi-spheres), and so we can directly (and exactly) calculate the size of a minimal epsilon expansion.  On the cube, there are no known “geometric” results - the shapes that produce minimal expansion are unknown.  This is a widely discussed open problem (see, e.g., the top of page 11 in the journal article http://www.ugr.es/~aros/isoper.pdf). Fortunately, there are “algebraic” bounds on the size of an expansion that do not rely on the geometry of such sets.
>   It was not our intention to imply that we are the first to study such “algebraic” bounds, or that work has not been done in this area, but rather we were trying to explain what makes isoperimetric results on the cube less intuitive and more challenging than on the sphere.
>
> Thanks for pointing out the result by Ledoux.  While we were aware of this book on isoperimetric inequalities, we were not aware that it contained a result on the cube. We have updated the paper to make the origin of the result clear (2nd paragraph, Section 4).  We have decided to continue to include our version of the proof because Ledoux’s version produces much weaker constants than the fairly tight constants that we produce (this is not because our proof is superior in any way, but rather because Ledoux chose not to keep tight constants).  The paper contains an acknowledgement that our proof uses methods that appear in Ledoux's paper and earlier.
>
> We maintain that the proof of Theorem 5 is correct, but after looking back at the layout of the proof we understand the source of the reviewer’s confusion. Theorem 5 states a bound on the robustness of high-res classifiers, and then states a bound on low-res classifiers.  The original proof proved the statements in the opposite order (it proved the bound on low res classifiers and then high res).  We ask the reviewer to have a look at the revised version of the proof which has been re-ordered and clarified.
>
>    The proof of Theorem 5 is quite trivial (although we think non-obvious).  We think this theorem is valuable though, given that a number of papers now claim that high-res classifiers are inherently less robust than low-res classifiers.  Theorem 5 exhibits a simple class of imaging problems for which this is provably not so.
>
>
> Regarding the attribution of Lemma 4:  While we already included citations to the Milman, McDiarmid, and Talagrand in the original submission, we’ve updated the paper to include section numbers for these citations.
>
> Regarding image “complexity”:  We don’t think that further analysis can lend more strength to the interpretation of “complexity” here because it is just an interpretation and not a mathematical concept.  Our goal is just to give some intuition for what kinds of image sets have large/small U.
> We have made modifications to make clear to the reviewer that our use of the term "complexity" is informal and non-rigorous.  Also, see our comments about density estimation to the review above, which we think lends some strength to this interpretation.
>
> Regarding “meaningful vs active”:  we have revised this statement to make clear that the active bound may be quite large, and possibly not of interest (end of Section 6).

---

> > ### Comment · AnonReviewer1 · 2018-11-27
> > **wrt Theorem 5**
> >
> > (I'll read the updated draft and comment on other parts, right now I only comment on Theorem 5 related issues to get the discussion started.)
> >
> > After the clarification and seeing the new version of the paper, I still think Theorem 5 is problematic.
> >
> > What is the probability $p$ over? is it over models? data? or perturbation?
> > can you even define a probability measure on the model space? do the random images sampled from MNIST and b-MNIST need correspond to each other? do the perturbations in MNIST and b-MNIST need to correspond to each other?

---

> > > ### Author Response · Authors · 2018-12-05
> > > **We still maintain that Theorem 5 is correct, but we care about making sure the statement is clear**
> > >
> > > The Theorem begins with the statement "Suppose epsilon and p are such that, for all MNIST classifiers, a random image from class c has an epsilon-adversarial example (in the `2-norm) with probability at least p."
> > >
> > > To clarify this, the constant "p" satisfies the following condition:   choose any classifier C, and then choose an image randomly from the MNIST distribution.  With probability at least p (over the draw of the image) the randomly sampled data point has an adversarial example.
> > >
> > > Note: p must be chosen to be small enough that this condition holds *uniformly* over all classifiers.  In other words, the condition holds with the same p regardless of which classifier is chosen.  This is the meaning of the statement "for all MNIST classifiers."
> > >
> > > Here's the idea of the proof (informally):  Suppose L is the most robust classifier on the low-res dataset (in the sense that randomly sampled images have adversarial examples with probability p, for the smallest possible p).   Let H be the most robust classifier on the high-res dataset.  The theorem proves that the robustness of H cannot be worse than the robustness of L.  This is true because you can always create a high-res classifier by downsampling images and feeding them to the low-res classifier L.  By doing this, we create a classifier that achieves the robustness of L, but does it on the high-res dataset.  Since H is the most robust classifier, it's robustness needs to be at least as good as this multi-scale classifier we constructed.
> > >
> > > Note that this proof never constructs or names any particular perturbation.  However, an effective perturbation for the constructed high-res classifier could be down-sampled to make a perturbation for the low-res classifier. In this sense, the perturbation on these two classifiers would "correspond," as you say.  This correspondence does not invalidate the proof though.  This proof methods only requires us to construct 1 classifier with robustness p on the high-res dataset.
> > >
> > > Finally, regarding the statement "can you even define a probability measure on the model space?"  Our statement of the theorem assumes that MNIST images are sampled from a probability distribution.   We stated the Theorem for MNIST (rather than arbitrary data distributions) because we thought it made the result easier to state and easier to understand.  However, whether MNIST corresponds to a probability distribution is a philosophical issue that some might argue with.
> > >     We agree that this formulation could lead to the interpretation that the statement of the theorem is non-rigorous.  For this reason, we will change the statement of the theorem in the camera ready to be for an arbitrary image class sampled from the distribution function.

---

> > > > ### Comment · AnonReviewer1 · 2018-12-07
> > > > **wrt Thm 5 and others**
> > > >
> > > > Thank the authors for the detailed explanation. Now I believe Theorem 5 is correct.
> > > >
> > > > I suggest the authors make it very clear that p is over the randomly selected image, not over classes or over classification models, ideally in the theorem, or in a remark, or in the text. It is much easier for me to understand when I see "the most robust classifier… you can always create… " in the explanation, and this might help others too.
> > > >
> > > > I've also read other parts of the authors' response. I'll adjust my score to be 6.

---

### Official Review · AnonReviewer3 · 2018-11-05
**interesting maths; implications less clear**

**Rating:** 8
**Confidence:** 4

**Review:**

The paper considers the problem of adversarial examples in (mostly high-dimensional) multi-class classification problems. Although the results are not specific necessarily to very high dimensional data or two images, the paper mostly uses images as a running example, and so will I in the review.

Assume that the data all lies in the unit box in R^n ([0, 1]^n). A multiclass classifier with K classes partitions the unit cube into K parts, each part corresponding to a given class. There are distributions \rho_c associated with each class and there is a bound on their density given by U_c and the fraction of examples of class c is f_c. And (eps, p) adversarial point y for some point x is such that |x - y|_p <= \eps and the classifier classifies x & y differently.

The paper shows that under this modeling assumption adversarial examples are inevitable. The results mostly use standard (but deep) results from probability theory. The technical proofs themselves are not particular difficult (provided one has the right background). I think the overall implications are interesting, and I will recommend the paper be accepted.

However, I also feel that this is a missed opportunity. To some extent the authors do try to have some high-level discussion about adversarial examples, but I think this could be expanded on more. For instance, why should it be assumed that an example that is \eps far should automatically have the same class label? Surely, being "eps"-far away is an equivalence relation, thus this would mean that all the hypercube would have to be labeled by the same class. This is clearly not the case. One plausible explanation is that if you take two points that are in two different classes, then any sequence of points that take one to the other with the property that each adjacent pair is at most \eps far away, must have the property that some intermediate mass have negligible chance of being a "natural" image.

On the other hand, doesn't the fact that humans are not susceptible to most adversarial examples, imply that adversarial-example resistant classifiers exist? My own feeling is the assumption that U_c is bounded is the strongest assumption that may not hold true with real data. In any case, the paper has enough technical content to merit acceptance and I hope the open review forum will lead to a fruitful discussion about some of these questions.

--

Minor comments:
Page 6 (just after Thm 2). Isn't the bound in Eqn. (5) true for all \ell_p norms for p \geq 2? (not just \ell_2 as the sentence says)
Paras on Page 6 (just below Thm 2). It would be more pleasant if equation x could be replaced by Eq. (x) or Equation (x).
Para in Sec 7 on Unbounded density: Clarify what norm you mean when you talk about \eps/2 perturbations.
Thm 5: Seems odd to have a theorem about MNIST. Surely the result is a lot more general!!!

---

> ### Author Response · Authors · 2018-11-25
> **We thank the reviewer for taking the time to read our paper fully and provide careful comments**
>
> Regarding the epsilon-walk argument you discuss:  Your observation about passing through a region that does not contain “natural” images is correct.  One interesting thing about our theoretical framework is that a “class” is a distribution on the cube (the support of which might cover only a tiny fraction of the cube), while the “classifier” is a function that maps all points in the cube (not just the points that lie in the class distribution support) onto a label.  In our theory, it could be that two classes have distributions with supports that are separated by more than epsilon units.  However, it could still be easy to fool the classifier with an epsilon perturbation because the classifier assigns a label to all points, including things that don’t look natural.  There’s an argument to be made that this is what many classifiers do in real life;  adversarial examples might not lie on the “natural” image manifold because they contain “fuzz”  and other artifacts that natural images don’t, and yet they get assigned a label by the classifier.  One way to avoid this problem (at least in theory), which we discuss in the paper, is having a “don’t know” class.  In this case, one could degrade classifier performance by perturbing images into the “don’t know” class, but it might be difficult for an adversary to change the label to another defined class.  We have seen from the adversarial examples literature, though, that producing classifiers that don't assign strong labels to adversarially perturbed images might be easier said than done.
>
> Finally, we’ll say a few things about the reviewers comments on whether humans are subject to adversarial examples. There seems to be some debate about this.  It’s certainly true that, most of the time, attacks on neural nets don’t transfer to humans.  However, our experience has been that attacks on neural nets usually don’t transfer to other (black-box) neural nets either (although they sometimes do for certain pairs/ensembles of target/victim networks), and so we don’t think this observation conclusively resolves the issue of whether it’s possible to make adversarial attacks on humans.  To complicate things further, some authors claim to observe cases in which adversarial examples for neural nets do transfer to humans in certain contexts (https://arxiv.org/abs/1802.08195).   For what it’s worth, several neuroscientists and psychologist we have spoken to about this issue believe quite strongly that humans are susceptible to adversarial examples, just maybe not ones crafted using a neural net as a model for the human brain.
> We remain agnostic on this issue because it’s outside the scope of our expertise.   This question seems to lie in the realm of philosophy and psychology, and we’ve avoided it in our paper in favor of sticking to mathematical issues.
>
> Finally, thanks for pointing out a number of minor errors.  We have fixed them in the revision.  We agree with the reviewer that Eqn (5) is more clear than Eq 5, but unfortunately the non-parenthetical version seems to be the standard style chosen by the ICLR editors (they chose this unusual definition for the \eqref command).

---

> > ### Comment · AnonReviewer3 · 2018-11-26
> > **thanks for comments**
> >
> > As I was already positive about the paper, I won't say much. I think the added (non-mathematical) discussion about adversarial examples is useful. I also commend the authors for being reluctant to make unjustified claims outside their area of expertise---I didn't mean to suggest that they do so, but just would have liked to see a more substantial discussion, where if parts of it were more speculative, they would be clearly marked as being so.
> >
> > I would also like to recant my ridiculous claim that being \eps-close is an equivalence relation. Though, it is symmetric, and there is an \eps-path between any two points in the cube, thus the property that labels are same if points that are \eps-far away does lead to having all the cube being labelled by one class. As the authors have already understood what I meant to say and responded to it, no further comment is necessary.

---

### Official Review · AnonReviewer2 · 2018-11-07
**good insight on understanding adversarial examples**

**Rating:** 7
**Confidence:** 4

**Review:**

This paper uses several lemmas in geometry to prove that adversarial examples
are hard to avoid under the assumption that there is no "don't know" class and
the distribution of each class is not too concentrated. The paper first starts
with a simple case where the data points are distributed on a sphere, and then
extends the results to the realistic case where data points are inside a cube
[0,1]^n.

The paper uses epsilon expansion of a set as a mathematical tool, and borrows
some important lemmas from geometry to the case of adversarial learning.  In
the sphere case, the results come from a fact that high dimensional
half-spheres can almost cover all points in the sphere after an epsilon
expansion, and the results depend on dimension n. For the unit cube case, the
authors borrow a result from Talagrand, to show that the epsilon expansion of a
set can cover a large portion of the cube as long as the set distribution is
not very concentrated.  In this case, the results (for l_2 norm) do not depend
on dimension n.

Experimentally, the authors show that inputs with higher dimension can actually
get better robustness, aligning with the provided analysis.  The primary reason
that current adversarial defense does not work well on CIFAR is due to the fact
that dataset is more spread out in high dimensional space. This is a good
insight for understanding adversarial examples.

The paper is overall well written and easy to follow. The interpretation of
each lemma and proposition is clear. Although the paper mostly depend on
well-known results in geometry and the ideas used are simple, it does provide
good insight on explaining the prevalence of adversarial examples. I recommend
to accept this paper.

Question:
Is there any good method to estimate U_c for a dataset? Although it is intuitive
that CIFAR may have a smaller U_c than MNIST, is it possible to numerically
estimate this quantity? This is necessary to fully support the conclusions made
in experiments.

---

> ### Author Response · Authors · 2018-11-25
> **Thanks for the comments, and an answer to your question**
>
> We thank the reviewer for taking the time to carefully read our paper and provide feedback.  To answer the question:  Yes, there are methods for quantifying the density of CIFAR and MNIST, although the accuracy of these methods is disputed.  Classical density estimation methods (like Parzan windows and GMMs) fail on complex high-dimensional distributions.  However, neural-network-based methods can attack this problem by training a GAN on the dataset, and then using a formula that predicts the likelihood of samples produced by the GAN. This formula involves the Jacobian of the generator, and the density of the latent "z" that produced the image.  This was the approach taken in (https://arxiv.org/pdf/1705.08868.pdf).  The authors of that work use several different methods for training generative models, and find that the estimated densities are *highly* dependent on how the model is trained, although for each specific training method the predicted MNIST densities are much higher than CIFAR densities (see, e.g., Fig2 in the referenced arxiv paper).  This observation is compatible with the claims made in our paper.
> The issue of accurate density estimation on images is still an active area of research.   We have been collaborating with another lab to develop new methods for high-dimensional density estimation with the goal of getting more consistent results than previous methods.   We have omitted a discussion of these density estimates to remain anonymous (our work on density estimation is under review), but we will include a citation and a brief discussion in the camera ready.  To be transparent about our results, we find that typical CIFAR-10 images have log-densities roughly 40 orders of magnitude smaller than typical big-MNIST images, and these observed differences in density are compatible with the differences in adversarial robustness we observe for MNIST and CIFAR in Section 8.

---

> > ### Comment · AnonReviewer2 · 2018-11-26
> > **Thanks for the update**
> >
> > Dear Authors,
> >
> > Thanks for letting us know that you are working on estimating dataset densities. I am okay with the it if the AC wants to accept this paper, since this estimation problem is a challenging one on its own. Please make sure to add these references and further discussions in your final revision.
> >
> > Thanks,
> > Paper150 AnonReviewer2

---

### Public Comment · ~Weizhi_ZHU1 · 2018-10-10
**Any insights toward deep neural networks?**

Hi,

Thanks for your interesting paper.

However, your theory seems to be adaptive to all machine learning models rather than deep networks, am I right? Do you have further insights into why shallow learning doesn't suffer severe adversarial problems but deep learning does?

Another question, the right-hand side of (2) is dimension-free if you take p>=2, and becomes vol(A(\epsilon, d_p)) \geq \alpha + \sqrt{2\pi}\epsilon, which is only a little bit larger than vol(A). Can this bound support your argument, that adversarial examples are "everywhere"?

Thanks!

---

> ### Author Response · Authors · 2018-10-10
> **Deep nets vs shallow classifiers**
>
> Weizhi,
>   Thanks for taking the time to read and comment.  It seems that you’ve asked a number of different questions, so I’ve tried to address them each individually below.
>
> Is our theory neural-net specific?
> No.  Our theory is applicable to the general case of measurable classifiers.  We address the special case of neural nets experimentally, but not with analysis. Keep in mind that this enables us to address classifiers that we use in practice but are not pure neural nets.  For example, consider an adversarially hardened classification algorithm that first does median filtering, then JPEG compression, and then uses a neural classifier.  This pipeline is a measurable classifier (but not a neural net), and so our theory can say things about it.  We do think it’s interesting to study behaviors that are specific to neural nets, but that’s not what we did here.
>
> Why do deep nets seem more susceptible than linear classifiers?
> First, note that adversarially trained nets for MNIST are quite hard to fool without making severe changes to the image, and so it does not always appear that deep nets have poor robustness.  However, there are clearly datasets where the susceptibility of neural nets seems to be quite bad.
> There is a reason for this apparent susceptibility:  We *choose* to use neural networks on nasty datasets with very high “complexity”.  In Section 8, we show that it is fundamentally harder to avoid adversarial examples for complex datasets (e.g, ImageNet) than, say, a nice, linearly separable SVM dataset in which the data has a large margin and is highly concentrated near the corners/sides of the unit cube.
>   If you did use a linear classifier on ImageNet, it would be subjected to the same fundamental susceptibility bounds as a neural network.  Theorem 2 guarantees that, with some minimum probability, a random image is either (a) wrongly classified, or (b) correctly classified but with adversarial examples.   Linear classifiers for ImageNet are described by the former alternative (they’re wrong a lot), while neural nets pick the latter alternative (they’re usually correct, but have adversarial examples).
>
>
> Dimension-free for large p:
> There is an implicit dependence on dimensionality here that is easy to overlook. For large but fixed p (less than infinity - the infinite case is addressed in the paper at the end of Section 4), the radius of the unit cube goes to infinity as the dimension increases.  If one chooses epsilon to be proportional to the norm of a typical image, then epsilon increases in higher dimensions.  For this reason, if the concentration bound “U” remains fixed as the dimension increases, the theory still predicts an increase in adversarial susceptibility because of the increase in epsilon (even for large p).
> That being said, we show in Section 8 that there is not a fundamental link between adversarial robustness and dimensionality.  The shrinking of the exponential term in Theorem 2 is countered by a blow-up in the concentration bound ”U”.  As discussed above (and in Section 8), image complexity, and not dimensionality, is what affects the limits of adversarial susceptibility.
>
> Finally, I’d point out that we are not claiming that “adversarial examples are everywhere” for any one particular problem.  There are problems that are plagued by adversarial susceptibility, and problems that are not.  Rather, we are trying to take a rigorous look at what leads to adversarial susceptibility when it is present.

---

> > ### Public Comment · ~Weizhi_ZHU1 · 2018-10-15
> > **Thank you**
> >
> > Thanks very much for detailed and clear explanations.

---

### Public Comment · (anonymous) · 2018-11-05
**Excellent paper; but the "uniformity-over-dimension"-type of assumption should be more highlighted**

Thank you very much for this very interesting paper.
I strongly suggest accepting this paper (but I did not check the proofs).

Two remarks though:

1/ I think your paper (especially abstract and conclusion) should insist on the fact that your results necessarily imply adversarial examples (with high probability) only when the input distributions satisfy a kind of "uniformity-over-dimensions" assumption (which is captured by the fact that their density functions must be bounded, and this bound should not increase to quickly with the input dimension; see your very good discussions on b-MNIST). These uniformity-like implications of your assumptions should be clear to every reader, even if he only skims through the paper; especially since they are probably quite implausible for high-dimensional image data.
Btw: your results are very much in line (though more general) with those of [1], which should be cited: you both study adversarial examples when the input distributions are subject to some kind of "uniformity-over-dimensions" assumption.

2/ In the analysis of eq. (4), I suggest a short discussion (maybe instead of or after the paragraph following Theo 5; or appendix) on what happens when pasting in eq. (3)* of the paper [2], also submitted to this conference. This eq. (3) suggests to scale the epsilon attack-threshold in p-norm as d^{1/p}. (Note that for p=2, you get the same rate \sqrt{d} than in your own discussions after Theo 5.) When pasted into your eq. (4), the fraction's numerator reduces to exp(- \pi n) (when p \leq 2). It doesn't solve the case p = 0 (because of the denominator), but at least it shows that your discussions on the special case p=2 actually hold similarly for all 2 >= p > 0. That suggests that equivalent results probably hold also for the case p=0.
By the way, the results of [2] neatly explain your Figure 4a. More generally, it provides an alternative (or complementary) explanation of adversarial vulnerability to yours: rather than accusing the input distributions, it accuses the classifiers themselves by showing that, independently of the input distribution, our neural network "priors" (as implied by the network architecture and weight distribution at initialisation) yield too large gradients. It might be worthwhile to contrast both approaches/explanations of adversarial vulnerability in your paper (which incidentally brings us back to point 1/ ).

*sorry, in a first version I wrote eq. (5), but it's eq. (3) I meant
[1] Adversarial Spheres, Gilmer et al.
[2] Adversarial vulnerability of neural networks increases with input dimension, submitted to ICLR 2019

---

> ### Public Comment · (anonymous) · 2018-11-05
> **Biasing the reviewers**
>
> "I strongly suggest accepting this paper (but I did not check the proofs)."  from an anonymous poster definitely isn't helping the review process. That is a subjective opinion and this is not a social media forum. Maybe the ACs should discourage people from posting their personal opinions on accept/reject decisions here!

---

> > ### Public Comment · (anonymous) · 2018-11-06
> > **Maybe, but not sure. + Justification of my grading.**
> >
> > The goal of my comment was more about the two remarks than about the first two sentences.
> > But as I obviously read the paper in quite some detail (otherwise, how could I have made these two remarks?), and as I happen to know the field quite well myself, I thought I could as well give my opinion. Of course I want to bias the reviews, precisely because I think that the paper is very good. If everyone who seriously read the paper (those are  usually at least interested in the field, which cannot always be said of assigned reviewers) did the same and left a small appreciation, we would probably get a better picture of the value of the paper to the community, than through a very few neatly written but nevertheless very noisy complete reviews. (Of course, that assumes some code of good conduct, e.g. not posting about your friends; could be enforced by authorising only comments from accounts linked to some known institution and blocking all those from the domain of conflicts of the authors)
> >
> > That being said, the reason I wrote that this is an excellent paper is because I think that the overall question (are adversarial examples inevitable?) is highly relevant (yet too rarely mentioned in the literature), and that the authors provide valuable insights/contributions to its answer. Their results seem to stem from a basic learning-theory-like analysis and make some strong and probably not very realistic assumptions (boundedness of densities). But they have the merit to clearly identify and formulate mathematically the problem (which is already a big contribution), and to provide at least the start of an answer. Of course, it's not the end of the story; but almost no paper is.
> > As I was trying to explain with my remarks, my only little complaint is that it does not insist enough on  the greater picture (very useful for non-experts): is it our classifiers themselves that are biased or is it our data that cannot be classified? If it is the data, what are (on a high level) the assumptions that make it inherently  vulnerable (here: the "uniformity-over-dimensions"-kind of assumption). Concerning this second question, one could argue that everything one needs (the assumptions) is in the theorem. But those assumptions are not just technical: they are an essential part of the overall message, and therefore should be mentioned even in high-level explanations.
> >
> > Nevertheless, I hope that the paper will get accepted and not dismissed for reasons like "we don't know whether this description really applies  to real-world models". If we don't know, then it's a valid hypothesis and even more a reason to accept the paper. The community will have to show in future if this paper was right or wrong, and in doing so, it will inevitably sharpen its understanding of the phenomenon. That's how natural sciences work.

---

> ### Author Response · Authors · 2018-11-25
> **Thanks for the comments - we've noted them in our revisions**
>
> In response to your remarks:
>
> 1)  The question of whether adversarial examples are inevitable is an ill-posed one. Clearly, any classification problem has a fundamental limit on robustness to adversarial attacks that cannot be escaped by any classifier.   However,  these limits depend not only on fundamental properties of the dataset, but also on the strength of the adversary and the metric used to measure perturbations.  The purpose of this paper is to characterize the relationship between these quantities and adversarial robustness.
>    Your first comment pointed out to us that we need to make these subtleties clear in the paper so that we don't mislead a casual reader/skimmer.  We have made some changes to the intro, and major changes to the conclusion to make this clear.  We will likely make some further changes to the intro to clarify the exact assumptions we make, although right now we're packed for space so we need to think a bit on how to best do this.
>   That being said, we don't think that "uniformity-over-dimensions" quite captures that kind of assumptions we make - in fact, we deliberately avoid assuming that densities remain constant over dimensions.  The purpose of Section 8 is to take a rigorous look at how the density bound U_c changes with dimensions, and to show that high dimensionality *does not* inherently lead to adversarial susceptibility (Theorem 5).
>
> 2)  Thanks for the reference.  We have added a citation to this paper, and a brief explanation of its results, in our literature review.  The paper mostly addresses the case of un-training networks with random weights, but this may help explain what we call “accidental susceptibility,” i.e., weakness in the classifier that results from flaws in the training process.  We address the fundamental limits of susceptibility, which is a somewhat different angle on the problem, but we think that both types of susceptibility are important and  it’s worth looking at the issue from both angles.

---

### Public Comment · (anonymous) · 2020-02-03
**Counter example to Theorem 2  and suggestion of correction**


The article deals with an important topic and the set-up of the theorems is very interesting.
But unfortunately the result of Theorem 2 is not correct at least for $\epsilon<1$.

Consider the following counter example to Thm. 2:
Let $m=2$, $b\in (0,1/2)$ and $\rho_1$ be uniformly distributed on $[0,b]\times[0,1]^{n-1}$ and $\rho_2$ be uniformly distributed on $(b,1]\times[0,1]^{n-1}$. Let $C$ be a perfect classifier. Then $U_1=\frac{1}{b}$ and $f_1=b<=1/2$.
Let $x$ be a random point sampled from $\rho_1$.
Then, the probability of a misclassification or an adversarial example in distance smaller than $\epsilon$ is
\begin{equation*}
\mathbb{P}(x\in [b-\epsilon,b]\times[0,1]^{n-1})=\epsilon \,\text{vol}([b-\epsilon,b]\times[0,1]^{n-1})= \frac{\epsilon}{b} \to 0 \quad(\text{as } \epsilon\to 0)  \tag{$\ast$}
\end{equation*}
but the lower bound from Thm. 2 is e.g. for $p=2$
\[
1- \frac{1}{b}\, \frac{\text{exp}(-\pi\epsilon^2)}{2\pi} \to 1- \frac{1}{b2\pi} \quad (\text{as } \epsilon \to 0)
\]
Now choose $b=1/2$, then, the lower bound from Thm. 2 converges to $1-\frac{1}{\pi}=0.618$ as $\epsilon\to 0$, which contradicts $(\ast)$.

In my opinion the correct lower bound in Thm. 2 is
\[
1-U_c\, \frac{\text{exp}(-\pi n^{1-2/p^*} \epsilon^2)}{2\pi n^{1/2-1/p*} \epsilon},
\]
i.e. with an additional factor $\frac{1}{\epsilon}$ in the second summand.

The problem in the proof is in the derivation of the simplified formula in Lemma 3, which uses an upper bound for the cdf of the normal distribution.
This upper bound is not applied correctly. A correct application leads to an additional factor $1/\epsilon$ in the second summand of (3). This should lead to corrections in the results based on this inequality in Thm. 2 and the following discussion.

---

> ### Author Response · Authors · 2020-02-03
> **You are correct - we posted a fix to arxiv**
>
> You are correct!  We made a mistake applying the last equation in the proof of lemma 3.  Theorem 2 of our paper (as it appears on OpenReview) states that an $\ell_2$ adversarial example exists with probability at least
> $$1-U_c \exp(-\pi  \epsilon^2  )/(2\pi ).$$
> This bound is incorrect.  The correct bound is
> $$1-U_c \exp(-2\pi  \epsilon^2  )/(2\pi \epsilon).$$
> We have uploaded an amended version to arxiv where this correction has been made.
>
> Note, the new (correct) bound is actually quite a bit stronger than the (incorrect) one that appears in open review.  Not only has a factor of 2 appeared in the exponent (which brings the probability bound closer to 1), but also the factor of $\epsilon$ in the denominator tightens the bound in cases where $\epsilon>1.$  In high dimensions, we expect larger epsilons to be used, and the adversarial examples presented in the intro were created with $\epsilon=10.$  However, for very small epsilons and in low dimensions (like the ones used in the counter-example above), the corrected bound is weaker.

---

### Meta-Review · Area_Chair1 · 2018-12-16
**Interesting contribution to our understanding of adversarial examples**

**Confidence:** 5
**Recommendation:** Accept (Poster)

**Metareview:**

There's precious little work asking existential questions about adversarial examples, and so this work is most welcome. The work connects with deep results in probability to make simple and transparent claims about the inevitability of adversarial examples under some assumptions. The authors have addressed the key criticisms of the authors around clarity.